# Discrimination of CpG Methylation Status and Nucleotide Differences in Tissue Specimen DNA by Oligoribonucleotide Interference-PCR

**DOI:** 10.3390/ijms21145119

**Published:** 2020-07-20

**Authors:** Takeshi Shimizu, Toshitsugu Fujita, Sakie Fukushi, Yuri Horino, Hodaka Fujii

**Affiliations:** Department of Biochemistry and Genome Biology, Hirosaki University Graduate School of Medicine, 5 Zaifu-cho, Hirosaki, Aomori 036-8562, Japan; munchsan@hirosaki-u.ac.jp (T.S.); h16m1091@hirosaki-u.ac.jp (S.F.); h17m1090@hirosaki-u.ac.jp (Y.H.)

**Keywords:** ORNi-PCR, PCR, acetone-fixed paraffin-embedded (AFPE), formalin-fixed paraffin-embedded (FFPE), bisulfite, CpG methylation, polymorphism, mutation, epidermal growth factor receptor (EGFR)

## Abstract

Oligoribonucleotide (ORN) interference-PCR (ORNi-PCR) is a method in which PCR amplification of a target sequence is inhibited in a sequence-specific manner by the hybridization of an ORN with the target sequence. Previously, we reported that ORNi-PCR could detect nucleotide mutations in DNA purified from cultured cancer cell lines or genome-edited cells. In this study, we investigated whether ORNi-PCR can discriminate nucleotide differences and CpG methylation status in damaged DNA, such as tissue specimen DNA and bisulfite-treated DNA. First, we showed that ORNi-PCR could discriminate nucleotide differences in DNA extracted from acetone-fixed paraffin-embedded rat liver specimens or formalin-fixed paraffin-embedded human specimens. Rat whole blood specimens were compatible with ORNi-PCR for the same purpose. Next, we showed that ORNi-PCR could discriminate CpG methylation status in bisulfite-treated DNA. These results demonstrate that ORNi-PCR can discriminate nucleotide differences and CpG methylation status in multiple types of DNA samples. Thus, ORNi-PCR is potentially useful in a wide range of fields, including molecular biology and medical diagnosis.

## 1. Introduction

PCR is widely used throughout biology and medicine [1,2], and PCR-based detection of nucleotide mutations has been used in clinical diagnoses of intractable diseases, such as cancer [3]. In this context, PCR with a specific primer set can specifically detect a target mutation. Alternatively, PCR can amplify a sequence across a target mutation, followed by DNA sequencing of the amplicon to confirm the presence of the mutation.

Nucleotide mutations can also be detected by blocking PCR [4], in which a blocker oligonucleotide hybridized to a target sequence suppresses PCR amplification across the target sequence in a sequence-specific manner. If the target sequence is mutated, hybridization is incomplete, and PCR amplification would not be suppressed. By monitoring DNA amplification, one can determine the presence or absence of mutations in a target sequence. Previously, we developed a form of blocking PCR, called oligoribonucleotide (ORN) interference-PCR (ORNi-PCR) [5], in which an ORN serves as the sequence-specific blocker, as shown in Figure 1. ORNs can be flexibly designed and cost-effectively synthesized, representing advantages over other blockers, such as artificial nucleic acids. We showed that ORNi-PCR can discriminate differences in nucleotide sequences, such as single-nucleotide mutations in cancer cells, as well as indel mutations introduced by genome-editing [6,7,8,9]. Thus, ORNi-PCR is a useful tool for use in the biological and medical fields.

In previous work, we used DNA extracted from cultured cell lines or complementary DNA (cDNA) reverse-transcribed from freshly prepared RNA for ORNi-PCR [6,7,8,9]. Such DNA is generally of high quality and can readily be used as a template for ORNi-PCR or conventional PCR. By contrast, DNA extracted from fixed tissue specimens is generally damaged and unsuitable for PCR, as fixation procedures using formalin and organic solvents, such as alcohol and acetone, cause DNA fragmentation [10,11,12,13]. On the other hand, although bisulfite-treated DNA has been widely used for PCR to distinguish the state of CpG methylation, the bisulfite treatment that converts cytosine to uracil also damages DNA and causes difficulty with PCR amplification [14,15]. Considering the potential use of ORNi-PCR for clinical diagnosis, it is important to determine whether such damaged DNA is compatible with ORNi-PCR.

In this study, we conducted feasibility studies to determine whether tissue specimen DNA and bisulfite-treated DNA were compatible with ORNi-PCR. We showed that ORNi-PCR could discriminate nucleotide differences in DNA extracted from acetone-fixed paraffin-embedded (AFPE) specimens and formalin-fixed paraffin-embedded (FFPE) specimens. In addition, we showed that ORNi-PCR could discriminate CpG methylation status in a target sequence using bisulfite-treated DNA. Our results suggest the potential utility of ORNi-PCR for a variety of applications in biological and medical fields.

## 2. Results

### 2.1. ORNi-PCR Using AFPE or FFPE Specimen DNA

Formalin has been widely used to preserve tissue specimens for a long time because it fixes tissues more strongly than other organic solvents. However, organic solvents, including acetone, has also been employed to fix tissues for some cases, in which milder fixation was suitable for downstream analyses, such as immunohistochemistry with specific antibodies [16,17]. Indeed, we have used acetone fixation for the immunohistochemistry of rat tissues [18]. Thus, we first examined whether DNA extracted from AFPE tissue specimens could be used for ORNi-PCR. Previously, we reported a nucleotide polymorphism in the rat *Gstm1* gene (encoding glutathione S-transferases mu 1); Sprague–Dawley (SD) rats have the wild-type (WT) sequence (^198^Lys (AAG)-^199^Ser (AGC)) (considering nucleotides G and A, we hereafter call this type of nucleotides as “GA-type”), whereas Hirosaki hairless rats have a nucleotide polymorphism (^198^Asn (AAT)-^199^Cys (TGC)) (considering nucleotides T and T, we hereafter call this type of nucleotides as “TT-type”), as shown in Figure 2A [19]. In this experiment, we investigated whether ORNi-PCR could discriminate the polymorphism in DNA extracted from AFPE rat liver specimens. To this end, we designed ORN_GA2 and ORN_GA3, 20-base and 22-base ORNs, complementary to the GA-type sequence, respectively, as shown in Figure 2A. DNA was extracted from sliced AFPE rat liver specimens and used for ORNi-PCR, as described previously [7]. As shown in Figure 2B,C, ORNi-PCR with ORN_GA3 suppressed the amplification of the GA-type but not TT-type *Gstm1*. By contrast, the amplification of the control region was not affected by the addition of ORN_GA3, as shown in Figure 2A,C, confirming sequence-specific suppression by the ORN. ORNi-PCR with ORN_GA2 yielded similar results, as shown in Appendix A. These results demonstrated that ORNi-PCR can discriminate nucleotide differences in AFPE specimen DNA.

We next examined the sensitivity of ORNi-PCR using rat AFPE specimen DNA. To this end, we mixed DNA including the GA-type and TT-type *Gstm1* alleles. ORNi-PCR amplified the *Gstm1* gene even when the TT-type *Gstm1* allele accounted for 0.5% of total *Gstm1*, as shown in Figure 2D. DNA sequencing analysis of the amplicon (#1) yielded only the sequencing signal for the TT-type *Gstm1* allele, as shown in Figure 2E. Thus, ORNi-PCR can detect a nucleotide difference in AFPE specimen DNA, so long as >0.5% of the corresponding sequence includes that difference. This sensitivity was not significantly different from that of ORNi-PCR using DNA purified from frozen rat livers, as shown in Appendix A.

Next, we performed ORNi-PCR on DNA extracted from FFPE specimens. To this end, we purchased two FFPE specimens mimicking patient samples: #HD141, consisting of a human cell line with the WT epidermal growth factor receptor (*EGFR*) gene; #HD300, consisting of a human cell line with the WT *EGFR* gene and five human cell lines possessing various mutations in *EGFR* corresponding to EGFR G719S, T790M, ΔE746–A750, L858R, and L861Q. The FFPE specimen #HD300 was assembled so that each mutant *EGFR* accounted for 5% of all *EGFR* genes. We extracted DNA from the FFPE specimens and investigated whether ORNi-PCR could detect mutated *EGFR* DNA, corresponding to EGFR L858R (CTG → CGG), as shown in Figure 3A. As shown in Figure 3A–D, ORNi-PCR with ORN_EGFR_L858 [7] successfully detected the mutated *EGFR,* even when it accounted for only 5% of total *EGFR*. By contrast, ORN_EGFR_L858 did not affect the amplification of the irrelevant *GAPDH* gene, as shown in Figure 3E. Thus, ORNi-PCR can detect a nucleotide difference in FFPE specimen DNA, so long as >5% of the corresponding sequence includes the difference. We previously studied the sensitivity of ORNi-PCR with genomic DNA (gDNA), purified from cultured cell lines using the same method as employed in Figure 2D [7]. In our previous study, ORNi-PCR detected the mutation corresponding to EGFR L858R when the mutated *EGFR* accounted for >0.1% of the total *EGFR*. Thus, the sensitivity of ORNi-PCR with FFPE specimen DNA was ~30-fold lower than that of ORNi-PCR with DNA purified from cultured cell lines [7], probably because formalin fixation strongly and covalently fixes samples [10,11,12,13], which would severely damage DNA. Notably, under the experimental condition, shown in Figure 3B, we did not detect the mutation corresponding to EGFR L861Q (CTG → CAG), although ORN_EGFR_L858 could also target the corresponding WT sequence, as shown in Appendix A. However, after optimization of the temperature of ORNi-PCR, we succeeded in detecting both mutations, as shown in Appendix A.

### 2.2. ORNi-PCR Using Whole Blood Specimens

Whole blood specimens are widely used for clinical diagnosis. If whole blood specimens could be directly used for ORNi-PCR without DNA extraction, the range of use of ORNi-PCR for clinical diagnosis would be markedly extended. Therefore, we attempted to use rat whole blood as a source of templates for ORNi-PCR. To this end, we collected whole blood from an SD rat (GA-type *Gstm1*) and a Hirosaki hairless rat (TT-type *Gstm1*) and subjected the samples directly to ORNi-PCR to discriminate the *Gstm1* polymorphism. In this experiment, we employed KOD FX (Toyobo) for ORNi-PCR because this reagent can more efficiently and stably amplify target DNA from crude samples, such as blood and tissue lysate [20]. As shown in Figure 4, the amplification of the GA-type *Gstm1* allele was suppressed in the presence of ORN_GA2, as shown in Figure 2A, whereas that of a control region was barely affected. These results suggest that whole blood specimens can be directly used for ORNi-PCR without DNA extraction.

### 2.3. ORNi-PCR Using Bisulfite-Treated DNA

Next, we investigated whether ORNi-PCR is compatible with bisulfite-treated DNA. Bisulfite treatment converts unmethylated cytosines to uracils but leaves methylated cytosines unchanged, as shown in Figure 5. Bisulfite treatment followed by PCR with a methylation-specific primer set (methylation-specific PCR (MSP)) has been used to discriminate CpG methylation status [21,22]. PCR with a universal primer set, followed by DNA sequencing analysis, has also been used to analyze CpG methylation status at single-nucleotide resolution [22]. Analysis of CpG methylation is also useful in the medical field (e.g., in cancer diagnosis [23]). In this experiment, we sought to determine whether ORNi-PCR can discriminate CpG methylation status in a target region of bisulfite-treated DNA, as shown in Figure 5. For this purpose, we employed a mutant version of the KOD DNA polymerase (KOD -Multi & Epi-^TM^ (Toyobo)) for ORNi-PCR because this polymerase effectively amplified a target region from bisulfite-treated DNA, as shown in Appendix A. We designed two ORNs targeting a CpG island of the human *CDKN2A (p16)* gene, ORN_p16_U and ORN_p16_M, targeting unmethylated and methylated CpG sites, respectively, as shown in Figure 6A,B and Appendix A. To evaluate those ORNs, we performed ORNi-PCR on plasmids pMD20_p16_M and pMD20_p16_U; pMD20_p16_U retains the DNA sequence corresponding to the bisulfite-treated unmethylated CpG island, whereas pMD20_p16_M retains the sequence corresponding to the bisulfite-treated methylated CpG island, as shown in Figure 6C (see the Materials and Methods section). As shown in Figure 6D,E, ORN_p16_U strongly suppressed amplification with pMD20_p16_U but not pMD20_p16_M. By contrast, ORN_p16_M suppressed amplification with pMD20_p16_M but not pMD20_p16_U, although the suppression was moderate. A standard concentration of ORNs (i.e., 1 μM) [7] was less effective, as shown in Appendix A.

We next applied established experimental conditions, as shown in Figure 6D, to ORNi-PCR using commercially available bisulfite-treated human gDNA (fully CpG-methylated and unmethylated gDNA) or bisulfite-treated gDNA purified from a human colorectal carcinoma cell line, HCT116. In HCT116 cells, the *CDKN2A (p16)* genes are CpG-methylated in one allele but not in the other allele (i.e., there exist one copy of CpG-methylated and another copy of unmethylated *CDKN2A (p16)* gene) [24]. ORNi-PCR with ORN_p16_U and ORN_p16_M strongly suppressed the amplification of the target *CDKN2A (p16)* sequence from bisulfite-treated unmethylated and CpG-methylated human gDNA, respectively, as shown in Figure 7A. Amplification from bisulfite-treated HCT116 gDNA was moderately suppressed by each ORN, probably due to the hemi-allelic CpG methylation of *CDKN2A (p16)*. To confirm the suppression events with bisulfite-treated, HCT116 gDNA, PCR, and ORNi-PCR amplicons were subjected to DNA sequencing analysis, as shown in Figure 7B,C. Sequencing signals for both CpG-methylated and unmethylated sequences were detected equally in the absence of ORNs, as depicted by #1 in Figure 7C. By contrast, sequencing signals of the CpG-methylated and unmethylated sequences were predominant in the presence of ORN_p16_U and ORN_p16_M, respectively, as shown in #3 and #4 in Figure 7C. When ORNi-PCR was performed with bisulfite-treated 293T gDNA, target amplification was completely suppressed in the presence of ORN_p16_U, as shown in Figure 7B (upper). This is reasonable because the *CDKN2A (p16)* gene is completely unmethylated in 293T cells, shown by #2 in Figure 7C. Together, these results demonstrated that ORNi-PCR can discriminate CpG methylation status in a target region in bisulfite-treated DNA.

Finally, we sought to determine the sensitivity of ORNi-PCR for bisulfite-treated DNA. To this end, we mixed gDNA of HCT116 (one copy of each CpG-methylated and unmethylated *CDKN2A (p16)*) and 293T (two copies of unmethylated *CDKN2A (p16)*), so that the CpG-methylated *CDKN2A (p16)* accounted for 5–0.05% of total *CDKN2A (p16)*, and then subjected them to bisulfite treatment. We used ORN_p16_U because of its stronger suppressive effects, as shown in Figure 6E and Figure 7C. As shown in Figure 7D, ORNi-PCR with ORN_p16_U amplified the target sequences only when CpG-methylated *CDKN2A (p16)* accounted for 5% of total *CDKN2A (p16)*. DNA sequencing analysis of the amplicon yielded sequencing signals only from the CpG-methylated sequence, as shown in Figure 7E. Thus, ORNi-PCR can discriminate the status of CpG methylation so long as 5% of target CpG sites are CpG-methylated. This sensitivity is not high relative to other blocking PCR methods [25,26]; however, the evaluation protocols are not comparable (details are provided in the Discussion section).

## 3. Discussion

In this study, we conducted various feasibility studies of ORNi-PCR with different kinds of DNA. We showed that DNA extracted from AFPE and FFPE specimens and bisulfite-treated DNA were compatible with ORNi-PCR. First, we showed that, in addition to DNA extracted from cultured cell lines, DNA extracted from freshly prepared tissues and AFPE and FFPE specimens can be used for ORNi-PCR. A general concentration (0.5–2 μM) of ORNs [7] is compatible with these template DNAs, as shown in Figure 2 and Figure 3 and Appendix A. Therefore, the optimal concentrations of ORNs for ORNi-PCR with these template DNAs do not considerably differ. However, the sensitivity of ORNi-PCR is lower with FFPE specimen DNA.

We found that whole blood specimens could be directly used for ORNi-PCR without DNA extraction. We used ORN_GA2 for ORNi-PCR with whole blood specimens, as shown in Figure 4 because we could not determine an optimized condition for ORNi-PCR with ORN_GA3 that suppressed GA-type amplification using whole blood specimens. ORN_GA3 is two bases longer than ORN_GA2, as shown in Figure 2A. Therefore, we speculated that a part of ORN_GA3, including the longer sequence, might bind to similar DNA or RNA sequences in a non-specific manner, which might reduce the frequency with which ORN_GA3 binds to the target sequence and prevent the suppression of target amplification. Alternatively, whole blood contains various molecules other than DNA (e.g., RNA, proteins, sugar, and lipids), and these may inhibit ORN/DNA hybridization. Considering that unknown factors may affect ORNi-PCR, more trial-and-error approaches may be required to find an optimal experimental condition for ORNi-PCR with whole blood specimens. Nevertheless, if an optimal experimental condition could be established, this method would be more straightforward to detect nucleotide differences in the DNA of blood cells that are abundant in whole blood (e.g., lymphoid cells). On the other hand, the detection of nucleotide differences in cell-free DNA (cfDNA) and circulating tumor DNA (ctDNA) has received attention in medical fields, such as cancer diagnosis [27]. Future studies should investigate whether cfDNA and ctDNA can be used for ORNi-PCR, although in such cases, DNA separation steps would be unavoidable. Next-generation sequencing (NGS) panels can be applied to identify candidate driver mutations more comprehensively. Multiplex ORNs designed for targeted NGS panels may help to detect such mutations more sensitively.

A general concentration (0.5–2 μM) of ORNs [7] was not effective for ORNi-PCR using plasmid DNA or bisulfite-treated DNA, as shown in Figure 6 and Figure 7. There are several possible explanations for this finding. First, we previously defined the general concentration of ORNs using gDNA purified from cultured cell lines. In general, target sequences can be amplified more easily from plasmid DNA than gDNA. Such a difference in template DNA may affect the optimal concentrations of an ORN. Second, we previously defined the general concentration (0.5–2 μM) of ORNs using KOD -Plus- Ver. 2 (Toyobo) [7]. In ORNi-PCR using bisulfite-treated DNA, a mutant version of the KOD DNA polymerase (KOD -Multi & Epi-^TM^ (Toyobo)) was employed. Such a difference in DNA polymerases may affect the optimal concentration of an ORN. Third, in ORNi-PCR using bisulfite-converted DNA, ORNs hybridize with their target sequences from the second PCR cycle, as shown in Figure 5 and Appendix A. By contrast, ORNs hybridize with their target sequences from the first PCR cycle in ORNi-PCR using DNA without bisulfite treatment, as shown in Figure 1. Such different modes of hybridization may affect the optimal concentrations of ORNs. Fourth, the hybridization of DNA/ORN (ORN_p16_U or ORN_p16_M) might be weaker than that with other ORNs tested so far. Thus, the optimal concentrations of ORNs may vary depending on the types of template DNAs, DNA polymerases, or hybridization modes. If a general concentration (0.5–2 μM) of ORNs is not effective, higher concentrations of ORNs should be considered. On the other hand, ORN_p16_U suppressed target amplification more effectively than ORN_p16_M in ORNi-PCR with bisulfite-treated DNA, as shown in Figure 6 and Figure 7. ORN_p16_U is 1 base longer, as shown in Figure 5, and may therefore hybridize with the target sequence more strongly than ORN_p16_M, resulting in the stronger suppression of target amplification.

Various methods to detect CpG methylation have been developed. Considering their advantages and disadvantages [23], a method that is suitable for the objectives should be selected. In addition to these methods, we showed that ORNi-PCR can discriminate the CpG methylation status, so long as 5% of target CpG sites are methylated, as shown in Figure 7. In this regard, another blocking PCR method, HeavyMethyl, can detect 30 pg of bisulfite-treated SssI-methylated DNA in 50 ng of bisulfite-treated unmethylated DNA (detection of <0.1% CpG methylation), although it uses an artificially methylated template [25]. HeavyMethyl employs two 3′-phosphorylated DNAs as blockers to inhibit the annealing of forward and reverse primers in combination with real-time PCR using TaqMan probes. Owing to the double blocking DNAs and fluorescence detection system, this approach may detect CpG-methylated DNA specifically and sensitively. Another blocking PCR method, enhanced-ice-co-amplification at lower denaturation temperature-PCR (E-ice-COLD-PCR), amplifies bisulfite-treated CpG-methylated DNA in a sequence-specific manner [26]. E-ice-COLD-PCR also employs 3′-phosphorylated DNA, including some locked nucleic acids (LNAs) as a blocker to inhibit the annealing of a primer. E-ice-COLD-PCR, in combination with pyrosequencing, detected 0.5% of bisulfite-treated CpG-methylated DNA in bisulfite-treated unmethylated DNA [26]. In contrast with HeavyMethyl and E-ice-COLD-PCR, an ORN blocks the elongation of a DNA polymerase on a strand in ORNi-PCR. Although these blocking PCR methods show higher sensitivity than ORNi-PCR, they require specialized equipment, such as a real-time PCR thermal cycler and pyrosequencer. In addition, the syntheses of blockers other than ORNs, such as LNAs, can be expensive. By contrast, ORNi-PCR is cost-effective because it does not require such equipment. Therefore, if high sensitivity is not necessary, ORNi-PCR can be used to analyze the CpG methylation status more cost-effectively. Specificity and sensitivity may be improved if ORNi-PCR employs two ORNs complementary to both strands and a fluorescence detection system.

MSP has been widely used to discriminate the CpG methylation status [21,22]. It is a highly sensitive method (as little as 0.1% of methylated DNA can be detected) [21], but yields false positives due to incomplete bisulfite conversion or false priming in some cases [21,28]. ORNi-PCR using a methylation-specific primer set may be useful to reduce such false positives. Alternatively, when suitable primer sets for MSP cannot be designed, ORNi-PCR with universal primer sets can be used to discriminate the CpG methylation status. In addition, bisulfite sequencing has been used to assess the CpG methylation status at the single CpG-site level. ORNi-PCR can be used to enrich sequences where a specific GpG site is (un)methylated, which makes sequencing analysis more cost-effective. Alternatively, the sensitivity of MSP may be increased by using such enriched DNA.

We believe that ORNi-PCR is potentially useful in various fields, including molecular biology and medical applications. ORNs can be cost-effectively synthesized, meaning they can be used for various applications in a cost-effective manner compared with artificial DNA, such as LNAs and peptide nucleic acids. Future studies should evaluate ORNi-PCR on human clinical specimen DNA to assess its viability as a method for clinical diagnosis; however, further improvements to sensitivity may be necessary before the method can be applied in this context.

## 4. Materials and Methods

### 4.1. Oligonucleotides

Oligodeoxyribonucleotides (ODNs) were chemically synthesized (FASMAC, Tokyo, Japan) and used as primers. ORNs were chemically synthesized and purified by high-performance liquid chromatography (FASMAC, Tokyo, Japan). The ODNs and ORNs used in this study are listed in Appendix A.

### 4.2. Extraction of DNA From Rat Liver Specimens

SD rats and SD-derived Hirosaki hairless rats were maintained in the animal faculty of Hirosaki University Graduate School of Medicine. Rat livers were removed following the animal faculty guidelines. The livers were immediately stored at −80 °C or subjected to AFPE treatment as described previously [18]. DNA was extracted from frozen rat liver tissues (ca. 25 mg) using the DNeasy Blood & Tissue Kit (QIAGEN, Valencia, CA, USA) or AFPE rat liver tissues (20 µm slices) using the Quick-DNA FFPE Kit (Zymo Research, Irvine, CA, USA). After the measurement of its concentration, the DNA was diluted and used in each experiment. All animal experiments were approved by the Institutional Animal Care and Use Committee of Hirosaki University (#M17003, 28 February 2018; #M17005, 28 February 2018).

### 4.3. Extraction of DNA From FFPE Control Specimens

DNA was extracted from FFPE human specimens mimicking patient samples (20 µm slices), consisting of human cell lines possessing WT *EGFR* (HD141, Horizon Discovery, Cambridge, UK) or WT and mutant *EGFR* alleles (HD300, Horizon Discovery, Cambridge, UK), using the Quick-DNA FFPE Kit (Zymo Research, Irvine, CA, USA). After measurement of its concentration, the DNA was diluted and used in each experiment.

### 4.4. ORNi-PCR With AFPE or FFPE Specimen DNA

ORNi-PCR was performed using KOD -Plus- Ver. 2 (Toyobo, Osaka, Japan), as described previously [7]. Optimal conditions for ORNi-PCR were determined following guidelines described previously [7]. Briefly, KOD -Plus- Ver. 2, 20 ng of AFPE DNA (or DNA prepared from unfixed rat liver) or 10 ng of FFPE DNA, 0.3 μM each primer, and 0.5–2 μM ORNs were mixed in a total volume of 10 μL. Two-step ORNi-PCR was performed at various temperatures for the annealing and elongation steps in the presence of different concentrations of ORNs along with a method to determine the optimal ORNi-PCR conditions [7]. The established optimal conditions (concentrations of DNA and ORN, and temperatures) are shown in Figure 2B and Figure 3B.

ORNi-PCR products were electrophoresed on 2–3% agarose gels. If necessary, each product was subjected to DNA sequencing (Eurofins Genomics, Tokyo, Japan). DNA sequencing data were analyzed using the SnapGene^TM^ Viewer (GSL Biotech LLC, San Diego, CA, USA), a free software. The latest version of the software can be downloaded from a website of the company (https://www.snapgene.com/snapgene-viewer/).

### 4.5. ORNi-PCR With Whole Blood Specimens

Rat whole blood samples were acquired following animal faculty guidelines and used directly as templates for ORNi-PCR. ORNi-PCR was performed using KOD FX (Toyobo, Osaka, Japan) in mixtures containing 0.3 μL whole blood, 0.3 µM each primer, and 1 µM ORN in a total volume of 10 μL. ORNi-PCR was performed at various temperatures for the annealing and elongation steps in the presence of different concentrations of ORNs along with a method to determine the optimal conditions [7]. An established optimal condition is shown in Figure 4A.

### 4.6. Plasmid Construction

gDNA was extracted from 293T or HCT116 cells by a standard phenol/chloroform protocol. gDNA was subjected to bisulfite treatment using the EZ DNA Methylation-Direct^TM^ Kit (Zymo Research, Irvine, CA, USA). Bisulfite-treated HCT116 gDNA was subjected to PCR with TaKaRa EpiTaq^TM^ HS (for bisulfite-treated DNA) (Takara Bio, Shiga, Japan), with primers hCDKN2A-Bisul-CpG-free-F and hCDKN2A-Bisul-CpG-free-R, as shown in Appendix A, to amplify a CpG island of the human *CDKN2A* gene (encoding p16), which is completely CpG-methylated in one but not both alleles in HCT116 cells [24]. PCR cycles were as follows: 40 cycles of 98 °C for 10 s; 55 °C for 30 s; 72 °C for 1 min. PCR products were cloned into T-vector pMD20 (Takara Bio, Shiga, Japan) and subjected to DNA sequencing analysis. Plasmids containing CpG-methylated or unmethylated DNA-derived sequences were named pMD20_p16_M or pMD20_p16_U, respectively.

### 4.7. ORNi-PCR With Plasmid DNA or Bisulfite-Treated DNA

Plasmids pMD20_p16_M and pMD20_p16_U were used as templates for ORNi-PCR. Alternatively, EpiTect PCR Control DNA Set (QIAGEN, Valencia, CA, USA), bisulfite-treated 293T gDNA, and bisulfite-treated HCT116 gDNA were used as templates for ORNi-PCR. In this regard, the EpiTect PCR Control DNA Set (QIAGEN, Valencia, CA, USA) includes CpG-methylated and bisulfite-treated DNA, and unmethylated and bisulfite-treated DNA. gDNA was extracted from the 293T and HCT116 cell lines and subjected (500 ng of each) to bisulfite treatment using the EZ DNA Methylation-Direct^TM^ Kit (Zymo Research). Alternatively, 293T gDNA (500 ng) was mixed with HCT116 gDNA so that CpG-methylated *CDKN2A (p16)* accounted for 0–5% of the total *CDKN2A (p16)*, and then the mixture was subjected to bisulfite treatment. After the measurement of its concentration, bisulfite-treated DNA was diluted and used for ORNi-PCR. ORNi-PCR was performed using KOD -Multi & Epi-^TM^ (Toyobo, Osaka, Japan) in mixtures containing 100 fg plasmid DNA or 10 ng bisulfite-treated gDNA, 0.3 μM of each primer, and 1–6 μM ORN in a total volume of 10 μL. ORNi-PCR was performed at various temperatures for the annealing and elongation steps in the presence of different concentrations of ORNs along with a method to determine the optimal conditions [7]. An established optimal condition is shown in Figure 6D.

## Figures and Tables

**Figure 1 ijms-21-05119-f001:**
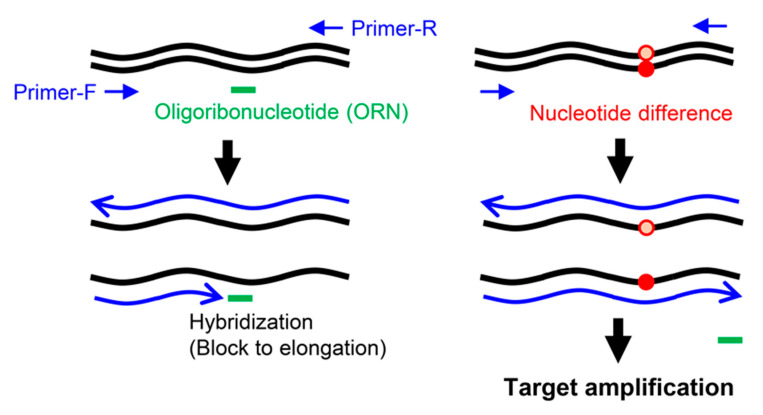
Schematic diagram of ORNi-PCR. In ORNi-PCR, an ORN hybridized with a target site blocks elongation (blue lines) of a DNA polymerase that does not possess 5′ to 3′ exonuclease activity. If the target site is mutated (red dots), the ORN cannot effectively hybridize to the target site, resulting in target amplification.

**Figure 2 ijms-21-05119-f002:**
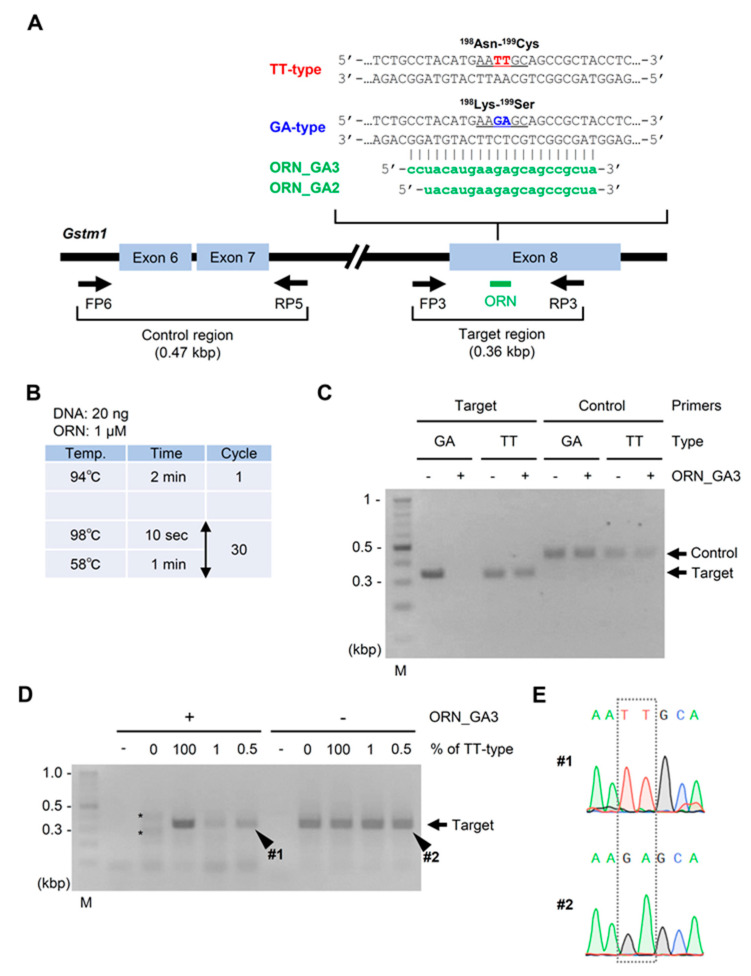
ORNi-PCR using DNA extracted from AFPE rat liver specimens to discriminate a polymorphism in the *Gstm1* gene. (**A**) Positions of primers and ORNs targeting the GA-type allele of the rat *Gstm1* gene. Arrows indicate forward and reverse primers for amplification of a target region (FP3 and RP3) and a control region (FP6 and RP5). (**B**) Experimental conditions for ORNi-PCR. (**C**) Results of ORNi-PCR. M—molecular weight marker. (**D**) Results of ORNi-PCR. ORNi-PCR with mixed DNA (0–100% of TT-type *Gstm1* in total *Gstm1*) was performed as shown in (**B**) with 40 cycles of denaturation and annealing/elongation steps. In this experiment, 20 ng of AFPE DNA, including GA-type *Gstm1*, was mixed with AFPE DNA, including TT-type *Gstm1*, so that the TT-type *Gstm1* accounted for 1 or 0.5% of the total *Gstm1*. The mixed DNA was used for ORNi-PCR. * indicates non-specific amplicons. (**E**) Results of DNA sequencing analysis. ORNi-PCR (#1) and PCR (#2) amplicons shown in (**D**) were subjected to DNA sequencing analysis. Sequencing signals around the polymorphism are shown.

**Figure 3 ijms-21-05119-f003:**
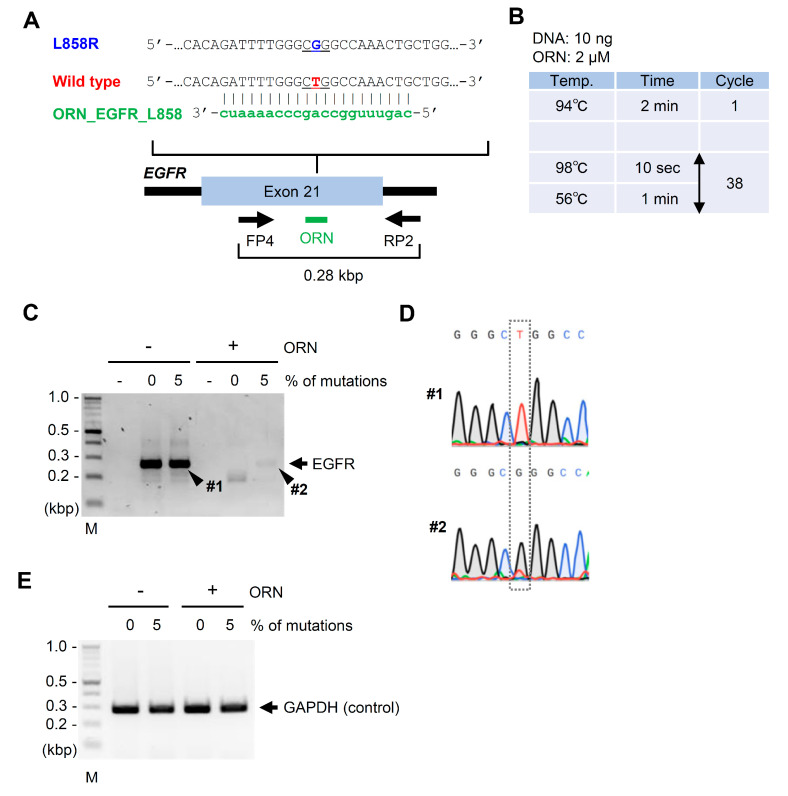
ORNi-PCR using DNA extracted from FFPE specimens to discriminate mutation of the *EGFR* gene. (**A**) Positions of primers and an ORN targeting a mutation corresponding to EGFR L858R in the human *EGFR* gene. Arrows indicate forward and reverse primers for amplification of the target region. (**B**) Experimental conditions for ORNi-PCR. (**C**) Results of ORNi-PCR. M—molecular weight marker. (**D**) Results of DNA sequencing analysis. ORNi-PCR (#2) and PCR (#1) amplicons shown in (**C**) were subjected to DNA sequencing analysis. Sequencing signals around the mutation are shown. (**E**) Results of ORNi-PCR targeting the irrelevant *GAPDH* gene.

**Figure 4 ijms-21-05119-f004:**
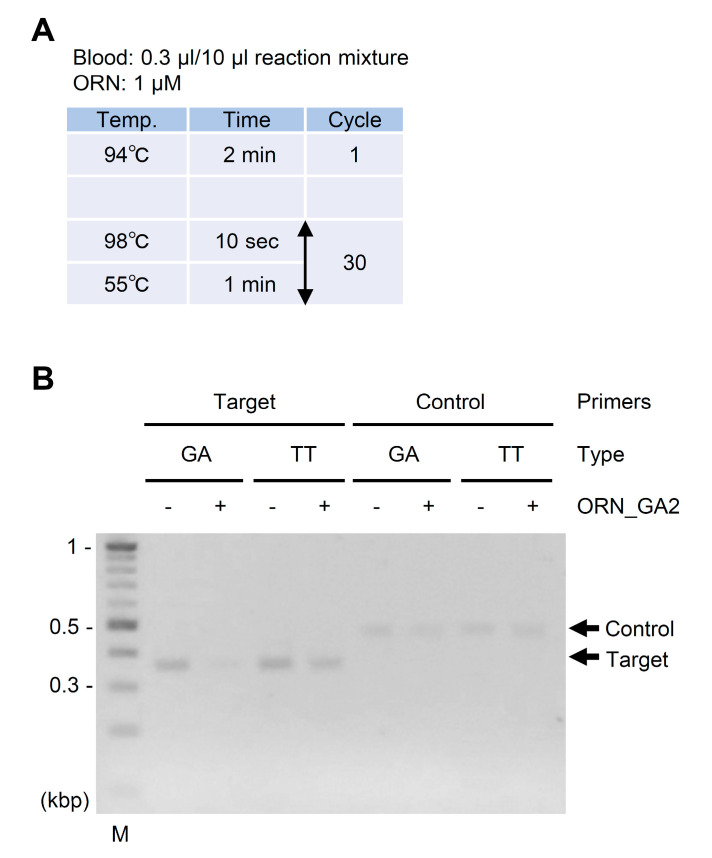
Discrimination of the *Gstm1* polymorphism in whole blood specimens by ORNi-PCR. (**A**) Experimental conditions for ORNi-PCR. (**B**) Results of ORNi-PCR. Rat whole blood was directly used as a template without DNA extraction. M—molecular weight marker. Positions of primers and ORN_GA2 are shown in Figure 2A.

**Figure 5 ijms-21-05119-f005:**
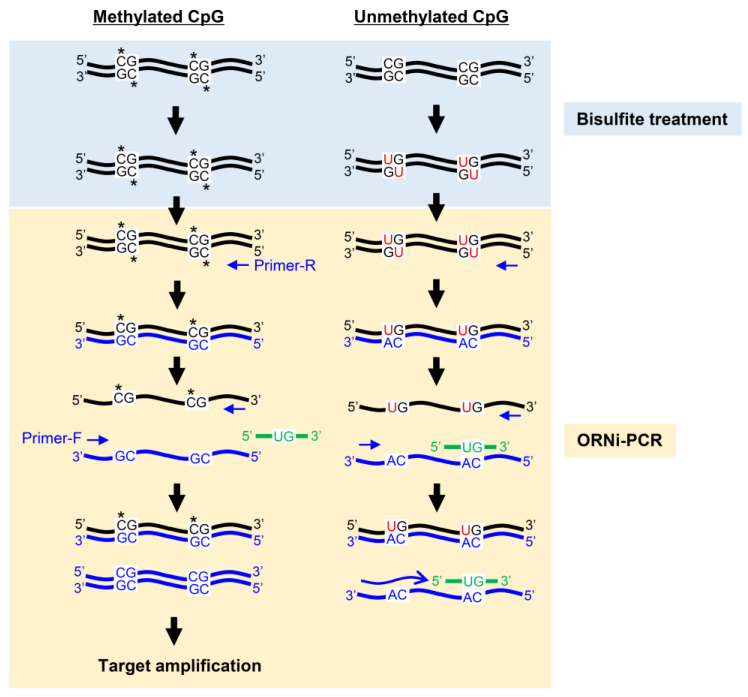
Schematic diagram of ORNi-PCR with bisulfite-treated DNA. Bisulfite-treated DNA is subjected to ORNi-PCR with an ORN corresponding to the converted target DNA sequence. In this scheme, an ORN (green line) targeting unmethylated CpG sites is hybridized with a corresponding DNA sequence synthesized in the first cycle of ORNi-PCR (blue line), which blocks target amplification.

**Figure 6 ijms-21-05119-f006:**
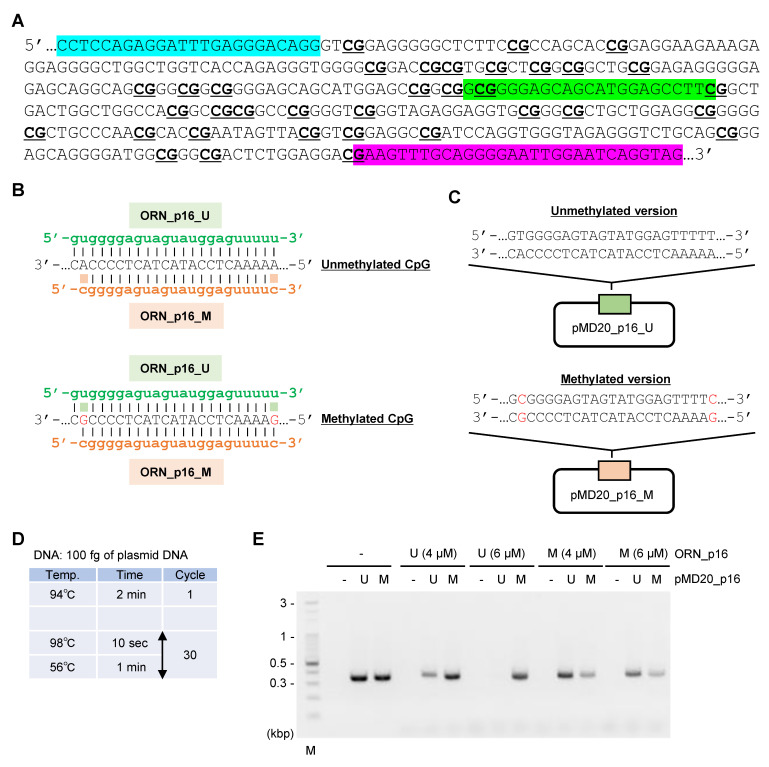
Design and evaluation of ORNs. (**A**) CpG island of the human *CDKN2A* gene (encoding p16). In HCT116, the CpG sites are completely methylated in one but not both alleles [24]. Primer positions are shown in light blue and pink. An ORN target site is shown in green. CpG sites are underlined. (**B**) ORNs designed for ORNi-PCR. The reverse DNA sequences synthesized in the first cycle of ORNi-PCR are shown, also depicted in Appendix A. (**C**) Plasmids used for evaluation of the designed ORNs. As to the nucleotide sequence corresponding to an ORN target site in pMD20_p16_M, nucleotides different from those in pMD20_p16_U are shown in red. (**D**) Experimental conditions for ORNi-PCR. (**E**) Results of ORNi-PCR. M—molecular weight marker.

**Figure 7 ijms-21-05119-f007:**
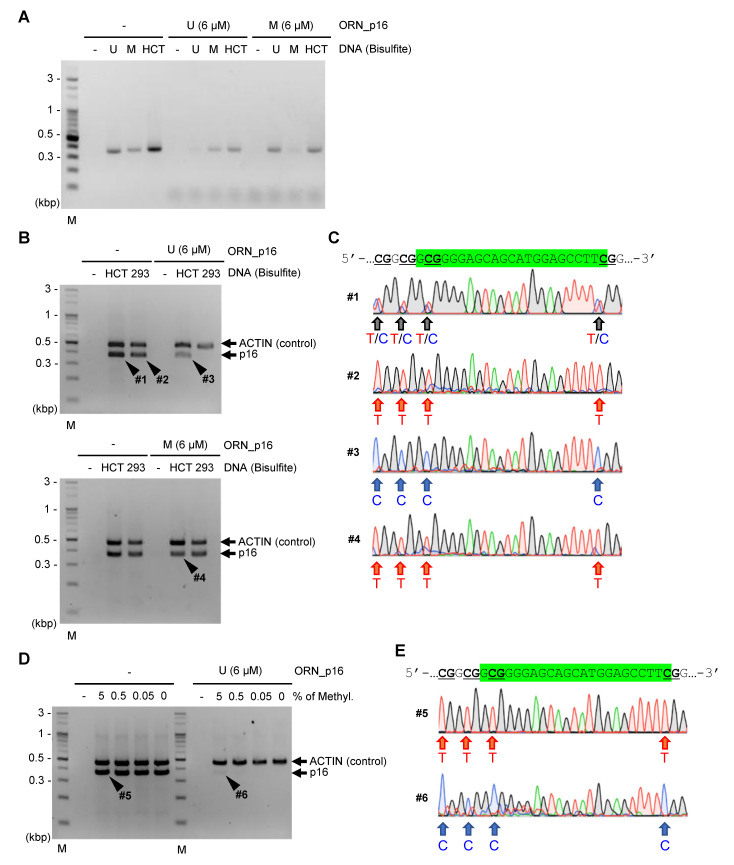
ORNi-PCR with bisulfite-treated DNA. (**A**) Results of ORNi-PCR with bisulfite-treated DNA. EpiTect PCR Control DNA Set (U and M) and bisulfite-treated HCT116 gDNA were used as templates. (**B**) Results of ORNi-PCR combined with internal control PCR. The human *ACTB* gene was amplified as an internal control. (**C**) Results of DNA sequencing analysis. PCR (#1, #2) or ORNi-PCR (#3, #4) amplicons, shown in (**B**), were subjected to DNA sequencing analysis. Sequencing signals around the ORN target site (green) are shown. CpG sites are underlined. (**D**) Sensitivity of ORNi-PCR with bisulfite-treated gDNA. 293T gDNA mixed with HCT116 gDNA (0–5% of CpG-methylated *CDKN2A (p16)* in total *CDKN2A (p16)*) was subjected to bisulfite treatment. (**E**) Results of DNA sequencing analysis. PCR (#5) or ORNi-PCR (#6) amplicons shown in (**D**) were subjected to DNA sequencing analysis. Sequencing signals around the ORN target site (green) are shown. CpG sites are underlined. ORNi-PCR was performed with 10 ng bisulfite-treated DNA, as shown in Figure 6D (**A**,**B**) or as shown in Figure 6D with 35 cycles of the denaturation and annealing/elongation steps (**D**).

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
