# Peer review of "Discrimination of CpG Methylation Status and Nucleotide Differences in Tissue Specimen DNA by Oligoribonucleotide Interference-PCR"

_ijms, 2020, doi:10.3390/ijms21145119_

Round 1

Reviewer 1 Report

This article is a follow-up of a previous study where the authors developed the oligoribonucleotide interference PCR (ORNi-PCR) methodology and demonstrated that this strategy could distinguish a single site mutation in cancer cell line models. In this follow-up, the authors assess whether the ORNi-PCR method can be extended to the analysis of single site mutations in DNA from fixed tissue samples (both acetone- and formalin-fixed paraffin embedded) and whole blood samples, or methylated DNA loci within bisulfate treated DNA. While in general, the authors demonstrate that ORNi-PCR can be applied to each of these sample types, which could have diagnostic potential, there are some aspects that should be included to strengthen the manuscript.

  • As stated above, while it appears that ORNi-PCR can generally be applied to all of the sample types evaluated, there are several places where the method does not seem to be as robust as expected. In these cases there is little to no discussion of these points. This is concerning in that it restricts the ability to evaluate the limitations of the ORNi-PCR method for various applications.

For example, within the whole blood samples, only the ORN_GA2 ORN worked to suppress target amplification, while this was not the case for the ORN_GA3. While this is commented on, no explanation is provided. Given that both ORNs work fine within other sample types, the possible reasons for this observation should be provided.

For the ORNi-PCR of bisulfite treated DNA depicted in Fig. 6D/E and SI Fig. S5, there is a clear concentration dependence relating the level of suppression to ORN amount, which is not at all discussed. It is also evident that the ORN for unmethylated DNA is much better at suppression than the equivalent methylated ORN, which is also not discussed. It is also not clear from the data in Fig. 7, why the results from bisulfite treated DNA from HCT116 and HEK293 cells are so different for the unmethylated ORN. Further, it is unclear if the cell line based DNA was bisulfite treated by the authors or commercially purchased. If the former, then details for this procedure should be included in the Materials and Methods section.

  • Another area in which the manuscript could be strengthened is by providing a more detailed comparison for each sample type how ORNi-PCR performs relative to the current state-of-the-art analysis method. For example, it is concluded that the sensitivity is not high for detection of mCpG over CpG relative to other blocking PCR methods, but no details are provided and the reader is left to wonder whether ORNi-PCR is a useful strategy for detection of this epigenetic mark. A more detail analysis of this aspect should be provided for each section in the Results and Discussion.

Author Response

Responses to Reviewer Comments

We thank the editor and reviewers for their helpful comments to improve the manuscript. We corrected the manuscript according to their suggestions. Revised wordings are shown in red in the revised document. Please find below our responses to specific issues raised by the reviewer #1.

Reviewer #1:

This article is a follow-up of a previous study where the authors developed the oligoribonucleotide interference PCR (ORNi-PCR) methodology and demonstrated that this strategy could distinguish a single site mutation in cancer cell line models. In this follow-up, the authors assess whether the ORNi-PCR method can be extended to the analysis of single site mutations in DNA from fixed tissue samples (both acetone- and formalin-fixed paraffin embedded) and whole blood samples, or methylated DNA loci within bisulfate treated DNA. While in general, the authors demonstrate that ORNi-PCR can be applied to each of these sample types, which could have diagnostic potential, there are some aspects that should be included to strengthen the manuscript.

We thank the reviewer for appreciating the potential utility of ORNi-PCR.

 1-1) As stated above, while it appears that ORNi-PCR can generally be applied to all of the sample types evaluated, there are several places where the method does not seem to be as robust as expected. In these cases there is little to no discussion of these points. This is concerning in that it restricts the ability to evaluate the limitations of the ORNi-PCR method for various applications.

We thank the reviewer for the insightful comment. In this regard, as mentioned by the reviewer #2, we separated the Result and Discussion section and described pros and cons on ORNi-PCR in detail in the new Discussion section.

1-2) For example, within the whole blood samples, only the ORN_GA2 ORN worked to suppress target amplification, while this was not the case for the ORN_GA3. While this is commented on, no explanation is provided. Given that both ORNs work fine within other sample types, the possible reasons for this observation should be provided.

We thank the reviewer for pointing out this issue. According to the reviewer’s comments, we added additional results of ORNi-PCR with AFPE DNA to show that ORN_GA2 is also effective to suppress amplification of the GA-type Gstm1 gene (new Supplementary Figure S1). Thus, Figure 2 and Supplementary Figure S1 clearly demonstrated that ORNi-PCR with ORN_GA2 or ORN_GA3 can suppress the GA-type Gstm1 gene from DNA purified from AFPE specimens. Even though, we could not determine an optimized condition for ORNi-PCR with ORN_GA3 that suppressed the GA-type amplificationusing whole blood specimens. ORN_GA3 is 2 bases longer than ORN_GA2. Therefore, we speculated that because a part of ORN_GA3 including the longer sequence might bind to similar DNA or RNA sequences in a non-specific manner, which might reduce the frequency with which ORN_GA3 binds to the target sequence and prevent suppression of target amplification. Alternatively, whole blood contains various molecules other than DNA (e.g., RNA, proteins, sugar, and lipids), and these may inhibit ORN/DNA hybridization. We added those description in the Discussion section. (Line 234-241)

 1-3) For the ORNi-PCR of bisulfite treated DNA depicted in Fig. 6D/E and SI Fig. S5, there is a clear concentration dependence relating the level of suppression to ORN amount, which is not at all discussed. It is also evident that the ORN for unmethylated DNA is much better at suppression than the equivalent methylated ORN, which is also not discussed.

We also thank the reviewer for pointing out these issues. We found that a general concentration (0.5–2 μM) of ORNs, which we reported in the reference #7, was not effective for ORNi-PCR using plasmid DNA or bisulfite-treated DNA (Figures 6 and 7). There are several possible explanations for this finding. First, we previously defined the general concentration of ORNs using genomic DNA purified from cultured cell lines. In general, target sequences can be amplified more easily from plasmid DNA than genomic DNA. Such a difference of template DNA may affect the optimal concentrations of an ORN. Second, we previously defined the general concentration of ORNs using KOD-Plus-Ver. 2. In ORNi-PCR using bisulfite-treated DNA, a mutant version of the KOD DNA polymerase (KOD Multi & Epi) was employed. Such a difference in DNA polymerases may affect the optimal concentration of an ORN. Third, in ORNi-PCR using bisulfite-converted DNA, ORNs hybridize with their target sequence from the second PCR cycle (Figure 5 and Supplementary Figure S5). By contrast, ORNs hybridize with their target sequences from the first PCR cycle in ORNi-PCR using DNA without bisulfite treatment (Figure 1). Such different modes of hybridization may affect the optimal concentrations of ORNs. Fourth, hybridization of DNA/ORN (ORN_p16_U or ORN_p16_M) might be weaker than that with other ORNs tested so far. Thus, optimal concentrations of ORNs may vary depending on the types of template DNAs, DNA polymerases, or hybridization modes. If a general concentration (0.5–2 μM) of ORNs is not effective, higher concentrations of ORNs should be considered.

 On the other hand, ORN_p16_U suppressed target amplification more effectively than ORN_p16_M in ORNi-PCR with bisulfite-treated DNA (Figures 6 and 7). ORN_p16_U is 1 base longer (Figure 5) and may therefore hybridize with the target sequence more strongly than ORN_p16_M, resulting in stronger suppression of target amplification.

 We added those description in the new Discussion section. (Line 252-271)

1-4) It is also not clear from the data in Fig. 7, why the results from bisulfite treated DNA from HCT116 and HEK293 cells are so different for the unmethylated ORN.

The CDKN2A (p16) gene is CpG-methylated in one but not both alleles in HCT116 cells. ORNi-PCR with ORN_p16_U suppressed amplification of bisulfite-treated unmethylated but not CpG-methylated the CDKN2A (p16) gene (Figure 7B and C, #1 vs #3). Therefore, the CDKN2A (p16) gene was still amplified by ORNi-PCR with ORN_p16_U. By contrast, the CDKN2A (p16) gene is completely unmethylated in 293T cells (Figure 7B and C, #2). Therefore, ORN_p16_U completely suppressed amplification of the CDKN2A (p16) gene from bisulfite-treated 293T genomic DNA. To avoid confusion by readers, we carefully re-wrote the corresponding parts. (Line 183-184, 189-192)

 1-5) Further, it is unclear if the cell line based DNA was bisulfite treated by the authors or commercially purchased. If the former, then details for this procedure should be included in the Materials and Methods section.

The commercially purchased DNA (EpiTect PCR Control DNA Set (QIAGEN)) is DNA products that have been already subjected to bisulfite treatment. However, as to HCT116 293T genomic DNA, we extracted genomic DNA from cultured those cells and then subjected them to bisulfite-treatment for the use of ORNi-PCR. To avoid confusion by readers, we added more information on bisulfite treatment in the Materials and Methods section.

 1-6) Another area in which the manuscript could be strengthened is by providing a more detailed comparison for each sample type how ORNi-PCR performs relative to the current state-of-the-art analysis method. For example, it is concluded that the sensitivity is not high for detection of mCpG over CpG relative to other blocking PCR methods, but no details are provided and the reader is left to wonder whether ORNi-PCR is a useful strategy for detection of this epigenetic mark. A more detail analysis of this aspect should be provided for each section in the Results and Discussion.

We thank the reviewer for the insightful comment. According to the reviewer’s comment, we described pros and cons on ORNi-PCR compared to other methods in detail in the new Discussion section.

Reviewer 2 Report

The presented manuscript describes the use of ORNi PCR for detection of mutations in fixed specimens as well as for detection of CpG island methylation in bisulfite treated samples. Overall the topic is interesting, but I have got lost in reading the manuscript. There are too many feasibility studies in the manuscript. There is not a clear description of the experiments. No data on DNA amount for each PCR was included, no description of the sensitivity studies. This renders the manuscript difficult to read and to evaluate. I appreciate if authors could re-organize the manuscript describing the results in a result section and commenting them in a separate discussion paragraph. Limit of detection of the method should be described carefully.

Some specific concerns

  1. Acetone fixation is not commonly used in histology for clinical tissues, why authors chose acetone fixed tissues? What is the rationale?
  2. Rows 109-110 This sensitivity was ~30-fold lower than that of ORNi-PCR with DNA purified from cultured cell lines [7], probably because formalin fixation strongly and covalently fixes samples [10–13], which would severely damage DNA. How was sensitivity studied? Please, describe the experiment.
  3. Rows 241-242: DNA was extracted from FFPE human specimens mimicking patient samples (20 μm slices), consisting of human cell lines possessing wild-type……… Have the authors tested also conventional FFPE specimens? In my experience, results from fixed cell lines can vary considerably with fixed tissues. I appreciate if authors include a standardization with scalar amount of DNA from FFPE fixed cells as well as FFPE tissues.
  4. Rows 247-248: Optimal conditions for ORNi-PCR were determined following guidelines described previously [7]. ORNi-PCR products were electrophoresed on 2–3% agarose gels. A brief description of the method should be included. Optimal ORN concentration with respect of the tissue type (e.g. fresh frozen vs fixed) should be described and discussed. The PCR protocol should be clearly written.

Author Response

Responses to Reviewer Comments

We thank the editor and reviewers for their helpful comments to improve the manuscript. We corrected the manuscript according to their suggestions. Revised wordings are shown in red in the revised document. Please find below our responses to specific issues raised by the reviewer #2.

Reviewer #2:

The presented manuscript describes the use of ORNi PCR for detection of mutations in fixed specimens as well as for detection of CpG island methylation in bisulfite treated samples. Overall the topic is interesting, but I have got lost in reading the manuscript. There are too many feasibility studies in the manuscript. There is not a clear description of the experiments. No data on DNA amount for each PCR was included, no description of the sensitivity studies. This renders the manuscript difficult to read and to evaluate. I appreciate if authors could re-organize the manuscript describing the results in a result section and commenting them in a separate discussion paragraph. Limit of detection of the method should be described carefully.

We thank the reviewer for the helpful comment. We previously established the methodology of ORNi-PCR and published their details (reference #7). In the present study, we performed various experiments based on the established methodology (reference #7). Therefore, to avoid redundancy of description, we simply cited the publication (reference #7) but did not show details on the materials and methods in this manuscript. However, as the reviewer mentioned, such lack of information may make readers confused. Thus, we added more technical information in the revised manuscript. In addition, according to the reviewer’s comment, we separated the Results and Discussion sections. Pros and cons on ORNi-PCR were discussed in detail in the new Discussion section.

2-1) Acetone fixation is not commonly used in histology for clinical tissues, why authors chose acetone fixed tissues? What is the rationale?

Formalin has been widely used to preserve tissue specimens for a long time because it fixes tissues more strongly than other organic solvents. However, formalin fixation is not suitable for some downstream analyses such as immunohistochemistry. Organic solvents including acetone can be employed in such cases (new references #16 and #17). Indeed, we have used acetone-fixation for immunohistochemistry of rat tissues (new reference #18). Thus, we first examined whether DNA extracted from AFPE tissue specimens could be used for ORNi-PCR. We added this information in Line 67-72.

 2-2) Rows 109-110 This sensitivity was ~30-fold lower than that of ORNi-PCR with DNA purified from cultured cell lines [7], probably because formalin fixation strongly and covalently fixes samples [10–13], which would severely damage DNA. How was sensitivity studied? Please, describe the experiment.

We previously studied the sensitivity of ORNi-PCR with genomic DNA purified from cultured cell lines using the same method as employed in Figure 2D (reference #7). In our previous study, ORNi-PCR detected the mutation corresponding to EGFR L858R when the mutated EGFR accounted for > 0.1% of the total EGFR. Considering our previous finding, we concluded that ORNi-PCR with FFPE specimen DNA showed lower sensitivity. We added this information in Line 114-117.

2-3) Rows 241-242: DNA was extracted from FFPE human specimens mimicking patient samples (20 μm slices), consisting of human cell lines possessing wild-type……… Have the authors tested also conventional FFPE specimens? In my experience, results from fixed cell lines can vary considerably with fixed tissues. I appreciate if authors include a standardization with scalar amount of DNA from FFPE fixed cells as well as FFPE tissues.

We thank the reviewer for pointing out this issue. In this regard, an aim of this study is to show that ORNi-PCR can be used with DNA extracted from human FFPE tissue specimens, especially clinical tissue specimens. Because we could not acquire clinical tissue specimens, we used commercially available FFPE tissue specimens mimicking human FFPE tissue specimens in this study. Some researchers may be interested in ORNi-PCR with DNA extracted from formalin-fixed cells. However, it would be out of scope in this study and be a future issue.

 2-4) Rows 247-248: Optimal conditions for ORNi-PCR were determined following guidelines described previously [7]. ORNi-PCR products were electrophoresed on 2–3% agarose gels. A brief description of the method should be included. Optimal ORN concentration with respect of the tissue type (e.g. fresh frozen vs fixed) should be described and discussed. The PCR protocol should be clearly written.

We thank the reviewer for the helpful comments. According to the reviewer’s comment, we described more technical information in the Materials and Methods section.

Round 2

Reviewer 2 Report

The authors responded properly to the questions I posed, however I suggest some minor mandatory corrections:

Rows 68-69: "However, formalin fixation is not suitable for some downstream analyses such as immunohistochemistry." I would like to stress the point that immunohistochemistry is a diagnostic method which is carried out in formalin fixed and paraffin embedded tissues, therefore I suggest deleting this sentence.

Rows 183-184: "The CDKN2A (p16) gene is CpG-methylated in one but not both alleles in HCT116 cells [24]." Please, re-phrase, the sentence is incomprehensible.

Row 257: "ORNs using KOD -Plus- Ver. 2." Please, define it in the discussion.

Please, check the abbreviations if defined in the manuscript at the first mention. 

Author Response

We thank the editor and reviewers for their helpful comments to improve the manuscript. We corrected the manuscript according to their suggestions. Revised wordings are shown in red in the revised document. Please find below our responses to specific issues raised by the reviewer #2.

Reviewer #2:

Rows 68-69: "However, formalin fixation is not suitable for some downstream analyses such as immunohistochemistry." I would like to stress the point that immunohistochemistry is a diagnostic method which is carried out in formalin fixed and paraffin embedded tissues, therefore I suggest deleting this sentence.

We deleted the sentence and additionally revised a related sentence (Line 68-70).

Rows 183-184: "The CDKN2A (p16) gene is CpG-methylated in one but not both alleles in HCT116 cells [24]." Please, re-phrase, the sentence is incomprehensible.

We re-phrased the sentence (Line 184-186).

Row 257: "ORNs using KOD -Plus- Ver. 2." Please, define it in the discussion.

We added the company information “(Toyobo)” after the original product name “KOD -Plus- Ver. 2”. In addition, we added the information of the general concentration of ORNs and reference #7 in the sentence (Line 259-261).

Please, check the abbreviations if defined in the manuscript at the first mention. 

We checked throughout the manuscript.